# Proximate Composition and Nutritional Values of Selected Wild Plants of the United Arab Emirates

**DOI:** 10.3390/molecules28031504

**Published:** 2023-02-03

**Authors:** Mohammad Shahid, Rakesh Kumar Singh, Sumitha Thushar

**Affiliations:** Division of Crop Diversification, International Center for Biosaline Agriculture, Dubai P.O. Box 14660, United Arab Emirates

**Keywords:** wild plants, amino acid content, fatty acid profile, mineral content, vitamin composition

## Abstract

Wild plants supply food and shelter to several organisms; they also act as important sources of many nutrients and pharmaceutical agents for mankind. These plants are widely used in traditional medicinal systems and folk medicines. The present study analyzed the nutritional and proximate composition of various compounds in selected wild plants available in the UAE, viz., *Chenopodium murale* L., *Dipterygium glaucum* Decne., *Heliotropium digynum* Asch. ex C.Chr., *Heliotropium kotschyi* Gürke., *Salsola imbricata* Forssk., *Tribulus pentandrus* Forssk., *Zygophyllum qatarense* Hadidi. The predominant amino acids detected in the plants were glycine, threonine, histidine, cysteine, proline, serine, and tyrosine; the highest quantities were observed in *H. digynum* and *T. pentandrus*. The major fatty acids present were long-chain saturated fatty acids; however, lauric acid was only present in *S*. *imbricata*. The presence of essential fatty acids such as oleic acid, α-Linoleic acid, and linolenic acid was observed in *H. digynum*, *S. imbricata*, and *H. kotschyi*. These plants also exhibited higher content of nutrients such as carbohydrates, proteins, fats, ash, and fiber. The predominant vitamins in the plants were vitamin B complex and vitamin C. *C. murale* had higher vitamin A, whereas vitamin B complex was seen in *T. pentandrus* and *D. glaucum*. The phosphorus and zinc content were high in *T. pentandrus*; the nitrogen, calcium, and potassium contents were high in *H. digynum*, and *D. glaucum*. Overall, these plants, especially *H. digynum* and *T. pentandrus* contain high amounts of nutritionally active compounds and important antioxidants including trace elements and vitamins. The results from the experiment provide an understanding of the nutritional composition of these desert plant species and can be better utilized as important agents for pharmacological drug discovery, food, and sustainable livestock production in the desert ecosystem.

## 1. Introduction

The body needs six nutrients for proper functioning and overall health. These include carbohydrates, proteins, fats, water, vitamins, and minerals. In many developing countries, starvation and undernourishment are on the rise because of population explosion, scarcity of productive land, and soaring food costs. The desert ecosystems in the Arabian region impose multiple abiotic stresses on vegetation, including chronic water scarcity, high temperatures and solar radiation, soil salinity, and low nutrient availability, which together limit plant growth and establishment. However, native desert flora has evolved physiological and morphological adaptations that enable persistence under these harsh conditions [1]. In the wild, there are many plants of high nutritional value that can be used to feed the ever-growing human population and help to secure nutritional security. Finding a diet from the wild has been closely related to humans for hundreds of thousands of years [2]. Around 12,000 years ago, before the invention of agriculture, people depended on Wild Edible Plants (WEPs) and hunting for survival. WEPs are found in a variety of botanical types, including herbs, vines, bushes, grasses, and trees, both annual and perennial species [3]. Approximately 30,000 edible plant species found around the world have the potential to be used as food or feed. This wild edible flora can potentially convert the food systems into more nourishing, sustainable, and resilient to abiotic stresses [4]. The WEPs provide both micro- and macronutrients that enrich the dietary quality, which are inexpensive sources of nutrition for different people worldwide [5,6]. Wild food use is still thriving, particularly in isolated, economically deprived places of the world [7,8] and during crop failures [9].

In addition to food and fodder, wild plants are also important sources of pharmacologically important agents such as antioxidant and anti-inflammatory compounds. Among these, the major ones include phytochemicals such as phenols, flavonoids, tannins, alkaloids, terpenoids, and saponins [10]. These compounds can break the reactive chain reactions such as free radicals and thereby inhibit oxidative insults to the body [11]. In addition, these compounds are known to inhibit enzymes and signaling cascades involved in inflammatory reactions [12,13]. Apart from these, phytochemicals are associated with the prevention and curing of various diseases such as cardiovascular diseases, neurodegenerative disorders, and various forms of cancers [14,15]. In addition to this, plants contain major vitamins such as vitamin B complex and vitamin C; these vitamins are central molecules involved in various physiological activities of the body and play a central role in redox balance [16,17]. Further, trace elements such as zinc, selenium, etc. are also known for their biologically important roles; considering their importance in the normal physiological functioning of the body and disease prevention, these plants are also considered to be important in medicine and food [18,19,20]. Hence, wild plants are important agents that provide various bioactive and pharmacologically important compounds to humans.

The United Arab Emirates (UAE), located in the southeast of the Arabian Peninsula, has an 83,600 square km area. The country has four major landforms: sand, salt flat, gravel, and mountains, which harbor a great diversity of flora that has adapted to diverse rough and mild environments found here. Most of the country is comprised of the desert, which is part of a vast sea of sand called Rub’ al Khali or Empty Quarter. Most plants here are either xerophytic (adapted to dry arid habitats) or halophytic (salt-tolerant). The UAE has about 820 plant species growing in different regions of the country [21]. Karim et al. [22] have screened 170 different plants in the UAE (trees, shrubs, and grasses) which are salt resistant along with their description, distribution, and uses. Native plants of the UAE were used in traditional medicine long ago in the UAE but in ethnobotanical literature most of the species found in the UAE are rare. Zayed Complex for Herbal Research and Traditional Medicine, a publication about medicinal plants that enlisted only 29 species, including some non-indigenous [23] species. Sakkir et al. [24] analyzed the medicinal status of UAE flora, and they listed a total of 132 plants (nearly 20% of total species) that are traditionally used in the UAE for their medicinal properties.

Wild plant species thus play a significant role in worldwide nutrition and food security [25,26] and the use of healthy edible wild flora can be investigated to support the nutritional/medicinal needs around the globe. A nutritional values database for WEPs needs to be systematically created to enhance dietary diversity and control hunger [10,20]. Information on wild plants’ edibility and medicinal properties is generally scarce, and data on their nutritional composition is insignificant [2,4,5,27]. In the present study, we aimed to evaluate the selected wild plants of the United Arab Emirates, viz., *Chenopodium murale*, *Dipterygium glaucum*, *Heliotropium digynum*, *Heliotropium kotschyi*, *Salsola imbricata*, *Tribulus pentandrus* and *Zygophyllum qatarense* which were found to grow abundantly in the UAE under the harsh environments for their nutritional and proximate composition.

*Tribulus* is characterized by buttercup-like yellow flowers and flora references of the region report five species in the United Arab Emirates (UAE), namely *T. macropterous* Boiss. (perennial), *T. omanense* (perennial), *T. parvispinos* Presl (annual), *T. pentandrus* Forssk. (annual/perennial) and *T. terrestris* L (annual, occasionally biennial). Among these, *T. pentandrus*, a well-distributed and less investigated species, were selected for the study. The native range of *T. pentandrus* (Arabic name: Zahar) belongs to the family Zygophyllaceae in the Arabian Peninsula. It is a perennial and grows primarily in the desert or dry shrubland biome(s) [28]. The native range of *C. murale* (Arabic name: Abu’ efein) belongs to the family Amaranthaceae in Macaronesia, Europe, Medit. to NE tropical Africa and Sri Lanka. It is an annual herb reaching 70 cm in height and grows primarily in the temperate biome(s) with an erect stem which is usually red or red-streaked green and leafy with green foliage. It has environmental uses, such as fodder, medicine, and food [29,30]. *D. glaucum* (Arabic name: Safrawi) belongs to the family Capparaceae and is very common in the UAE, along the Arabian gulf coast, often very close to beach lines, also on saline sand inland, except for the southern part of Abu Dhabi emirate. The plant prefers the deep sand of dunes and similar habitats where it can form fairly dense stands. Globally, the plant has been known from northern Sudan and Egypt east of the Nile through the Arabian Peninsula to the Desert areas of NW India in the provinces of Rajasthan, Gujarat, and Pakistan. This plant is much grazed by livestock and often very stunted, also a preferred feeding plant of the rare and endangered Houbara Bustard. The plant is used in some countries to treat respiratory diseases [31,32].

*Heliotropium*, with its different species, is considered a valuable medicinal plant worldwide. Genus *Heliotropium* belongs to Boraginaceae, a large family of dicotyledonous angiosperms which includes 16 genera, and 170 species present in the Mediterranean basin and the Middle East and extending through Europe and Tropical Africa [33]. The southwestern region of the Arabian Peninsula is considered a part of the floristic hotspot where there are many genera that have not received proper attention, e.g., *Heliotropium* L. [34]. The native range of *H. digynum* (Arabic name: Kary, Jery) is N. Africa to Iraq and the Arabian Peninsula. It is a perennial subshrub and grows primarily in the desert or dry shrubland biome(s). It is common in the UAE and widespread in sandy deserts, between dunes and sandy plains, and in the shade of Eucalyptus plantations. The plant is used potentially as a feed source for camels [35]. *H. kotschyi* (Arabic name: Ramram) is common and widespread along coasts; it is tolerant to salinity. By far it is the most common heliotrope in the UAE, abundant in all zones except steeper and higher mountains; it thrives in semi-saline flats and alluvial gravels and is not found in deep mobile sands. A poultice of leaves is used to treat blisters and snake bites [36,37,38]. *S. imbricata* (Arabic name: Ghadrib) belonging to the family Amaranthaceae is a perennial halophytic shrub that grows in deserts and arid regions of the Arabian Peninsula, southwestern Asia, and North Africa. *S. imbricata* can also be used as a model plant to study the cross-tolerance for salt and drought stress and improve the stress resistance in many other plant species [39]. The species has traditionally been used as a vermifuge and for treating certain skin disorders [40]. Five triterpene glycosides have been isolated from the roots of *S. imbricata,* with two of them being new glycoside derivatives not previously known [41]. It is used for producing alkali, eaten by camels only, with crushed leaves with a strong fishy smell and taste.

*Z. qatarens* (Arabic name: Hadidi) belonging to the family Zygophyllaceae is a salt-tolerant dwarf shrub with multiple stems that grows in the Arabian Peninsula and is both drought and salt tolerant. It has tiny, fleshy leaves with paired leaflets that are deciduous and store water, dropping off in stressful conditions, and can survive a leafless state for years [42,43]. It typically grows in coarse, stony, or sandy soils at the edge of salt flats, around salt marshes, and in the sand that accumulates on the base of depressions. It also grows in non-salty locations, on calcareous soils, and around the fringes of dune areas, and often dominates plant communities in these locations [44]. On well-drained sandy soil on coastal plains, it may cover 75% of the ground surface, and this plant community is probably the most encountered around the western side of the Persian Gulf [45]. The plants are found growing in association with several species of soil microfungi, regularly with *Cladosporium sphaerospermum*, but also sometimes with *Penicillium citrinum* and *Aspergillus fumigatus* [46]. Aqueous extract of the plant is documented to produce a lowering of blood pressure and acts as a diuretic and antipyretic, local anesthetic, with antihistamine activity, stimulation, and depression of isolated amphibian heart, relaxation of the isolated intestine, contraction of the uterus, and vasodilation. The extract antagonized acetylcholine action on skeletal muscle and acted additively to the muscle relaxant effect of d-tubocurarine. The juice from fresh leaves and stems is known to be used as an abrasive cleanser and as a remedy for the treatment of certain skin diseases. It is a rich source of water, and a treasure of salts made as food for camels [47].

Although little/no reports have been published on the nutritional value of these selected native plants from the UAE instead of *C. murale* [48], to contribute to the growing body of knowledge on this subject, we analyzed these seven plant specimens from the UAE for their lipid, fatty acid, protein, amino acid, and trace mineral content. It is expected that the results of the present study may yield important information regarding the utility of various plants present in the wilderness of the UAE.

## 2. Results and Discussion

### 2.1. Amino Acid Composition of Plants

The results revealed the presence of 17 various amino acids including eight essentials, in the aerial parts of the selected wild plant species (Table 1) from the United Arab Emirates (UAE). Different plant species exhibited varying concentrations of different essential and non-essential amino acids. *C. morale*, *D. glaucum*, *H. digynum*, and *T. pentandrus* contain all the 17 analyzed amino acids, while some amino acids are not detected in the selected plants due to their distribution profile being less optimal, viz., *H. kotschyi* (glutamic acid), *Z. qatarense* (histidine, cystine), and *S. imbricata* (glutamic acid, histidine, and cystine). The analysis shows that *H. digynum* has the maximum amount of amino acids (7.598 g/100 g) compared to the other six wild flora. The species contain higher amounts of glutamic acid, followed by aspartic acid, leucine, lysine, and alanine. For the amino acids’ concentration, the second in line is *T. pentandrus* (5.069 g/100 g), which is about 33% less than the first. On the other hand, *S. imbricata* has the lowest quantity of amino acids (1.21 g/100 g), which is six times less than *H. digynum*.

*H. digynum* also has the highest concentration (2.647 g/100 g) of essential amino acids (histidine, isoleucine, leucine, lysine, methionine, phenylalanine, threonine, tryptophan, and valine), followed by *D. glaucum* (1.898 g/100 g), with the difference of more than 28% between the two species. The lowest amount of essential amino acids was recorded in *Z. qatarense* (0.456 g/100 g). In all the plant species analyzed, tryptophan was unable to reach the limit of detection.

These amino acids are important in the normal functioning of the body and its regular activities. The amino acids that are normally not synthesized in the body are termed Essential Amino Acids (EAA); these plants act as the sources of various essential amino acids [49,50]. Further, the lower levels of these amino acids in the body can lead to complications such as metabolic diseases and other disorders [2,51,52]. EAA deficiency will lead to growth slowdown and development in children, the development of diseases, and the destruction of cells in adults. Among these amino acids, some are bioactive (lysine, isoleucine, leucine, valine, threonine, phenylalanine, and tyrosine), and others are antioxidants (histidine, methionine, and cysteine) [53]*. H. digynum* exhibited the highest glutamic acid content of 19.73%, and *S. imbricata* did not show a detectable amount. Glutamic acid acts as a neurotransmitter for the central nervous system, the brain, and the spinal cord. It supports the brain to correct the body’s physiological imbalances [54]. Serine is necessary for the muscles’ development and the immune system’s maintenance. It is important in the synthesis of RNA and DNA within the cells. Serine detected ranges from 4.1% to 7.7% in *Z. qatarense* and *T. pentandrus,* respectively. The values for alanine were between 4.8% (*S. imbricata*) and 5.6% (*Z. qatarense*). Alanine plays an important role in the transfer of nitrogen in the body and glucose that the body uses as energy and strengthens the immune system by producing antibodies. It also regulates the discharge of toxic substances. Proline ranges from 4.5% (*H. digynum*) to 6.3% (*T. pentandrus* & *Z. qatarense*). Proline will help to slow down the production of collagen thus facilitating improved skin texture and leading to the slowdown of the aging process. Proline plays an important role in curative treatment to avoid problems in the cartilage, tendons, and muscles of the heart [55]. Arginine values fluctuate between 4.7% (*H. digynum*) and 11.9% (*S. imbricata*). Arginine boosts the immune system thereby delaying the growth of cancerous tumors. Arginine plays a major role in the detoxification of the liver by neutralizing ammonia and reducing the toxicity of alcohol. In addition, it is frequently used in the treatment of infertility in men. Glycine hinders muscle degeneration; it improves glycogen storage and releases glucose to meet energy needs. Glycine values are ranges from 4.8% (*H. digynum*) to 6.6% (*T. pentandrus* and *S. imbricata*). Histidine is used in the treatment of cardiovascular disease with a physiological antioxidant role it plays on the free radicals (hydroxyl radical and singlet oxygen), and the values fluctuate from 1.9% to 3.9% in *T. pentandrus* and *C. murale*. Histidine cannot be detected in a traceable amount in *S. imbricata* & *Z. qatarense*. According to WHO, the daily need for this amino acid is 12 mg/kg or 840 mg to 70 kg of body weight. It is important to mention that methionine is an antioxidant with high sulfur content [25] that helps prevent deficiencies in hair cells, skin, and nails. Furthermore, this amino acid protects against greasy clusters around the liver and the arteries that cause obstructions. It promotes the detoxification of harmful agents such as lead and other heavy metals. Additionally, cysteine works as a powerful antioxidant to eliminate harmful toxins [56]. Therefore, these two-sulfur amino acid deficiencies would cause physiological disturbances, even the risks of contracting degenerative diseases [53]. Cysteine cannot be detected in *H. kotschyi*, *Z. qatarense*, and *S. imbricata*. All the wild plants except *S. imbricata* (8.0%) exhibited a range of 5% valine. Valine is used in the treatment of liver and gallbladder diseases and promotes brain vigor [56]. Threonine ranges from 4.8% (*H. digynum*) to 7.9% (*S. imbricata*). Threonine helps in maintaining the balanced intake of proteins in the body and is also an important part of the formation of dental enamel, collagen, and elastin. In the deficiency, the isoleucine causes physical and mental disorders. The highest range of leucine, isoleucine, valine, and phenylalanine was detected in *T. pentandrus* (9.5%), *Z. qatarense* (4.5%), *S. imbricata* (8.01%) and *C. murale* (7.2%). Leucine acts with isoleucine and valine to promote muscle, skin, and bone function [55]. In addition, it is recognized that the brain uses phenylalanine to produce norepinephrine, a chemical that transmits signals between nerve cells. Additionally, it promotes alertness and vitality, regulates human mood, and reduces pain. This amino acid is also used in the treatment of arthritis, depression, painful menstruation, migraine, obesity, Parkinson’s disease, and schizophrenia [55,56].

### 2.2. Fatty Acid Composition of Plants

The biochemical analysis of the leaves of seven wild species indicates the presence of 11 fatty acids in the present study (Table 2). All the plants contain C14:0 (myristic acid), C16:0 (palmitic acid), C18:0 (stearic acid), C18:1 (oleic acid), and C18:2 (linoleic acid) in their leaves. At the same time, C16:1 (palmitoleic acid) has been detected in *D. glaucum*, *H. digynum*, *S. imbricata*, *T. pentandrus*, and *Z. qatarense.* C17:0 (heptadecanoic acid) was reported in *C. murale*, *D. glaucum*, *H. kotschyi*, *T. pentandrus*, *Z. qatarense.* C18:3 *ɯ*3 (α-Linolenic acid) was in all plants except *C. murale.* On the other hand, C12:0 (lauric acid) was found in *S. imbricata*, and C20:1 (eicosenoic acid) in *H. kotschyi* only.

*T. pentandrus* had higher levels of palmitic acid and oleic acid; likewise, *H. digynum* contained higher amounts of palmitic acid, oleic acid, α-Linoleic acid, and linolenic acid. The predominant fatty acids in *D. glaucum* include palmitic acid, stearic acid, and oleic acid. In *S. imbricate*, palmitic acid, oleic acid, and α-Linoleic acid were the major fatty acids. *H. kotschyi* includes about 50% oleic acid as the primary fatty acid, whereas *Z. qatarense* and *C. murale* contain stearic acid, the primary fatty acid (50.4%).

The leaves of wild plants used for food have lower oil contents but are rich in essential fatty acids, e.g., linoleic acid (C18:2 *ɯ*6) and α-linolenic acid (C18:3 *ɯ*3) [57,58]. The human body cannot produce either α-linolenic acid or linoleic acid. These fatty acids are essential in regulating various metabolisms and have beneficial impacts against cardiovascular disease [27], non-alcoholic fatty liver disease, hyperlipidemia, and even cancers [59,60,61]. α-linolenic acid (ALA), an omega-3 amino, helps reduce inflammation in the human body. The results of the current study suggest that the leaf lipids of wild plant species, including the experiment (Table 2), are rich in essential fatty acids (18:2 *ɯ*6 and 18:3 *ɯ*3). Hence, the consumption of these plants may yield essential fatty acids.

The medium chain saturated fatty acids such as lauric acid are important in health and are known to have several pharmacological effects including anticancer properties [3,9,27] and hypolipidemic effects [4,5]. Oleic acid and α-Linoleic acid are important essential fatty acids, and plants act as important sources of these fatty acids. The essential fatty acids are important in the management of diseases including non-alcoholic fatty liver disease, hyperlipidemia and even cancers [6,62].

### 2.3. Nutrient Contents of Plants

The proximate composition of selected seven wild plants is described in Table 3. The predominant nutrients in different plants include carbohydrates, proteins, fats, ash, and fiber content. The highest level of carbohydrates, saturated fat, poly-unsaturated fats, fiber, and energy were present in *D. glaucum*. The highest protein content was observed in *H. digynum* (7.21 g/100 g) and the lowest in *C. murale* (3.09 g/100 g). The presence of high protein content suggests its nutritive superiority over other traditionally consumed crops. For total carbohydrates, *S. imbricata* (3.7 g/100 g) showed the least amount, while with 10.7 g/100 g, *D. glaucum* was at the top. The protein composition in the fresh leaves of the seven plants ranged from 1.75 to 7.21 g/100 g, and these values are close to the estimates of certain orphan leafy vegetables described in another report [25,63,64]. Diets that provide about 12% of their calorific amount from proteins are good protein sources [50]. Hence, these wild plants can have a significant role in supplying inexpensive and readily available proteins for people living in rural areas. The protein content in *C. murale* from Spain is reported to contain 4.35 ± 0.41 g [48], which is close to the values that we obtained in samples from the UAE. Plant proteins are cellular functional macromolecules that are required to perform a wide range of functions as enzymatic activities and managing transport across cellular membranes. The variation in protein content present in different species which belong to the same genus was previously reported [65].

The total fat content varied between 0.23 and 0.87 g/100 g in these plants, similar to the results of several studies that concluded that leafy vegetables are inferior resources of fats [4]. Though a sizable part of the fat present in the plant aerial parts is saturated, most of the lipids are unsaturated, either mono or poly. Unsaturated fats that are fluid at room temperature are useful as they can reduce blood cholesterols, relieve inflammation, soothe heartbeats, and play several other beneficial roles. Food supplying 1–2% of its caloric energy as fat is appropriate for humans, as excess fat intake results in cardiac diseases [50]. Fats are also a source of essential fatty acids such as linoleic and linolenic acid, which the body cannot synthesize and can only be acquired from diets. They are vital for managing inflammation, blood clotting, and brain function and development. The absorption of fat-soluble vitamins such as carotene and vitamin A in the body is also augmented by the presence of lipids [8].

The total carbohydrate content in the studied wild plant range was 3.5–22.7 g/100 g. There is a big difference (more than six times) for this important dietary compound among the seven species. A study in India [66] showed a lesser carbohydrate content in various green leafy vegetables consumed by some tribesmen there. Other research [67] also registered lower values for carbohydrate contents in wild plants eaten in eastern parts of India. On the other hand, the carbohydrates of the wild plant were comparable to those described by studies in Bangladesh [51], northeast India [68,69], and Pakistan [70]. The optimal daily carbohydrate needed for humans is 130 g. It indicated that 7.0–14.5% of the daily requirement could be achieved through the consumption of 100 g of these dried plants.

Ash contains all the important dietary ingredients, especially minerals, micro and macronutrients that are very significant for the normal physiological functions of the body. Ash comprises inorganic matter of the plant that contains oxides and salts, including anions, e.g., phosphates, sulfates, chlorides, and other cations and halides such as calcium, sodium, potassium, magnesium, manganese, and iron [8]. The ash content shows the aggregate of minerals in the food. This study found the highest ash in *Z. qatarense* (7.78 g/100 g) and the lowest in *D. glaucum* (3.5 g/100 g). The ash present in these wild plants is similar to some commonly used wild vegetables in Bangladesh, and India [51,68,69]. The ash content in *C. murale* collected from the UAE showed higher value as compared to the report from those collected from Spain (4.23 ± 0.28 g/100 mg) [48]. The quantity and composition of ash left over after burning of plant material varies significantly according to the plant’s age, time, organ to organ [71]. This must be evaluated further.

Moisture content is the amount of water present in a substance. Water is an important part of food. Around 20% of the total water intake is by diet moisture [32]. When foods are consumed, their moisture content is soaked up by the body. All the plant species under investigation had moisture content varying from 66.7% to 88.5% in fresh weight. The reasonably high moisture contents show that there might also be a higher activity of water-soluble enzymes [72]. The moisture content is determined mainly by the humidity, temperature, and harvest time of the plant species. The higher moisture contents in plants of less humid and dry regions might be due to their higher water retention capabilities because they have xeric nature and xerophytes store water and have sunken stomata to avoid transpiration of water. The maximum moisture content among the seven studied plants was in *C. murale* (88.5 g/100 g), and the lowest was in *D. glaucum* (66.7 g/100 g). *C. murale* having the highest moisture value makes it more prone to a decline in nutrients, since foods with high moisture content are more vulnerable to perishability. The present study also supports the above observation. The moisture content values of *C. murale* collected from the UAE in the present study is in correlation with samples collected from Spain (82.02 ± 3.01) [48].

Out of seven wild plants studied, the highest nutritive value regarding energy was noted in *D. glaucum* (124 Kcal/100 g) and the lowest in *C. murale* (22 Kcal/100 g). The energy was reported in the samples of *C. murale* (33 ± 19 Kcal) is comparable to the samples from the UAE used for the present study [48]. It has been reported that *D. glaucum* is used as herbal medicinal and camel’s favorite food source in the UAE. The camel prefers the plant especially when it is in flowering and seeding stages as a good source of revitalizing camel body. The tiny flowers of the plants work as a strong laxative agent. The stalk with flowers is a source of flashing the digestive system of the livestock as well as a human being. It is also given to camels suffering weakness and a lower desire for food [73]. Dietary diversification is important to improve the intake of critical nutrients. The low-fat content (0.54 to 2.37%) in fresh weight of aerial parts of the plants suggests that the plants can be used as a valuable source of good dietary practices and may be advised to individuals with overweight or obesity problems. Therefore, high protein, carbohydrate, and nutritional composition can form an ideal diet for children, breastfeeding mothers, and adults. Further toxicological studies need to perform the effective utilization of these plants as dietary supplements.

### 2.4. Vitamin and Mineral Composition

Vitamins are considered important nutrients in foods and carry out specific functions essential for health though their daily requirements are minute. The daily recommended intake of vitamin A, C, and riboflavin (B2) for pregnant women and children are (800 and 400 µg, 55 and 30 mg, 1.4 and 0.5 mg), respectively [74]. The predominant vitamins in the plants were the vitamin B complex. At the same time, a higher level of vitamin A was found in *C. murale*. The amounts of vitamin B2 and B3 were maximum in *T. pentandrus*, whereas vitamin B6 concentration was higher in *D. glaucum* compared to other species analyzed for the study (Table 4). B vitamins have an essential role in maintaining good well-being and directly influence energy levels, brainwork, and cell metabolism. Vitamin B complex may assist prevent microbial infections and help support cell health.

The essential minerals potassium (K), Calcium (Ca), and trace mineral zinc (Zn) of the seven wild plants are shown in Table 4 ranged between 438 and 9410 mg/kg in fresh plant leaves. D. glaucum shows the highest concentration, while Z. qatarense has the lowest amount; it plays a major role in our body to help in keeping normal levels of water within the cells. The Ca concentration varied between 1710 and 9662 mg/kg, while in some of the important leafy vegetables (cabbage, lettuce, and spinach), it varies between 390 and 730 mg/kg [8]. The outcome from this present study is roughly similar to the wild edible plants consumed in parts of rural Bangladesh [51] but better than the wild flora eaten as food in Pakistan [75]. Based on the finding, these wild plants may be a good Ca source for our food. This essential mineral is important for blood clotting and the normal working of the heart muscles [76].

Zn plays an important role in stabilizing macromolecular synthesis and structure. The metal ion’s role in the synthesis of the nucleotides (DNA and RNA) is well known, and both RNA and DNA polymerases are zinc-dependent enzymes. The highest Zn concentration was recorded for the studied seven plants in H. digynum (87.6 mg/kg) and the lowest in Z. qatarense (0.233 mg/kg). The results are close to the levels reported in some wild plants in India [77] and all the minerals quantified in C. murale from the UAE showed an increased amount as compared to samples from Spain [48].

Nitrogen (N) and phosphorus (P) are two main plant minerals, and their deficiencies frequently restrict floral development. Plants utilize N for leaf growth and its green color, while they use P to help form new roots and produce flowers, fruits, and seeds. The mineral also helps plants in fighting against diseases. The biochemical analysis of the plants reveals that H. digynum contains the highest amount of N (11,532 mg/kg), while Z. qatarense has the lowest (2514 mg/kg). For P, T. pentandrus has the maximum amount (1154 mg/kg), and with just 133 mg/kg, Z. qatarense is at the bottom. N is a vital part of the food that helps synthesize amino acids in the body. The amino acids are the building block of protein, and nitrogen is a part of their structure. P is an essential element of bone health and cellular activities in the body.

The micronutrients, e.g., trace elements and vitamins, are important in the normal functioning of the body [7,8,9]; they act as essential components of various enzymes including those regulate body antioxidant status [57,78]. In addition, these trace elements and vitamins act as inhibitors of oxidative stress and inflammation [25,26]; subsequently, these compounds prevent the development of various diseases including cancer and cardiovascular diseases [18]. Different habitats could directly affect the amounts of bioactive products accumulated in plants. Abd El Gawad et al. [79] compared phenolics content as well as the antioxidant activity of different Heliotropium species collected from coastal and inland habitats.

## 3. Materials and Methods

### 3.1. Collection and Preparation of Samples

Aerial parts of *C. murale* L., *D. glaucum* Decne., *H. digynum* Asch. ex C.Chr., *H. kotschyi* Gürke, *S. imbricata* Forssk., *T. pentandrus* Forssk., and *Z. qatarens* Hadidi were collected from the plants growing naturally at the ICBA campus (25.0947° N, 55.3899° E), Dubai, United Arab Emirates. All seven species are native to the United Arab Emirates (UAE) and are also found in various regions of Africa, Asia, and Europe (Table 5 and Figure 1). Other related information on the wild flora has been provided in the table. The samples were washed thoroughly 2–3 times with running tap water and then once with sterile water and used for further analysis. The detail of analysis is provided in Appendix A.

The soils at ICBA are sandy in texture, that is, fine sand (sand 98%, silt 1%, and clay 1%), calcareous (50–60% CaCO_3_ equivalents), porous (45% porosity). The saturation percentage of the soil is 26, with very high drainage capacity, while its saturated extract’s electrical conductivity (ECe) is 1.2 dS m^−1^. In line with American Soil Taxonomy [36], the soil is categorized as typic torripsamments, carbonatic, and hyperthermic [80].

### 3.2. Nutritional Analysis

Nutritional analyses followed the methodology of the Association of Official Analytical Chemicals [81]. Moisture content was determined by the difference between fresh matter and dry matter; fat was determined by the Soxhlet extraction method; protein (PS) was measured by the Kjeldahl method; crude fiber (CF) content was determined using the neutral detergent reagent method [82]; and total carbohydrate (CHO) content was estimated by the difference between 100 and the sum of the percentages of moisture, protein, total lipid and ash contents [83]. Total ash content (Ash) was analyzed after burning the plants in a muffle furnace. The micronutrients potassium (K^+^), calcium (Ca^2+^), phosphorus (P^3−^), iron (Fe^2+^) and zinc (Zn^2+^), were analyzed using the Inductive Coupled Plasma (ICP) spectrometer and atomic absorption [83].

### 3.3. Amino Acid and Vitamin Analysis

The amino acids were determined using Sykam Amino Acid Analyzer (Sykam GmbH, Eresing, Germany).

### 3.4. Fatty Acid Composition

The fat extracted from the samples were further analyzed for the fatty acids composition using gas–liquid chromatography (GLC) and gas chromatography mass spectrometry. More information about the methodologies applied and equipment used for the chemical analyses is provided in the Appendix A.

### 3.5. Trace Elemental Analysis and Proximate Composition

The trace element composition of the plant was estimated according to the standard methods. Briefly, the plant tissue was reflexed with concentrated nitric acid and perchloric acid; the samples were then analyzed by inductively coupled argon plasma atomic emission spectroscopy (ICP-AES). More information about the methodologies applied and equipment used for the chemical analyses is provided in the Appendix A.

## 4. Conclusions

A biochemical analysis of seven wild plants aims to unravel the rich dietary composition of indigenous plants and their capacity to be utilized as alternative sources of nutrients and nutritional supplements. The nutritional composition information of wild plant species will also help promote the use of more biodiverse foods and feed for healthy diets and the pharmaceutical industry in the UAE and elsewhere. The analyzed plant species could be an excellent substitute for other commonly eaten vegetables because of their superior nutrient content, and they should be tested for their toxicological properties. Unfortunately, many precious desert plants including the selected samples for the present study are depleting for numerous reasons. This flora really needs to be well studied, documented, and the status and risk factors ought to be identified. Most of these plants selected in the present study are used by the local population as fodder. However, no detailed study carried out an analysis of the nutritional quality of these plants for better utilization. Understanding the relative importance and preference of different species is also crucial for the sustainable management of the local forage resources and can help animal husbandry technicians to optimize the selection of useful fodder species and to improve the livestock system efficiency in the desert ecosystem.

## Figures and Tables

**Figure 1 molecules-28-01504-f001:**
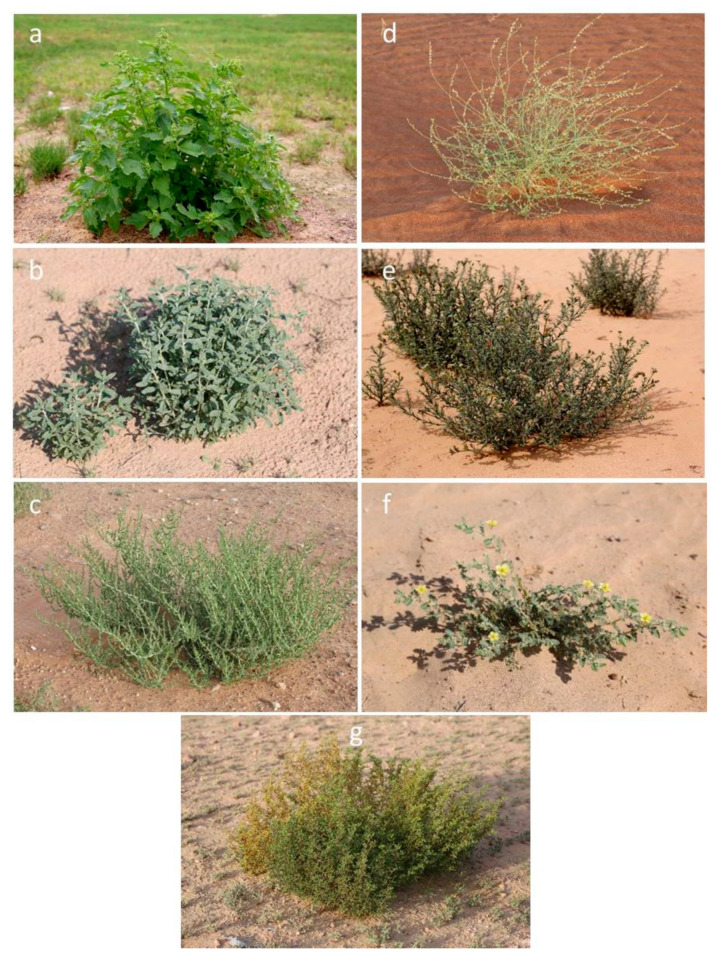
(**a**) *Chenopodium murale* L. (**b**) *Heliotropium digynum* Asch. ex C.Chr. (**c**) *Salsola imbricata* Forssk. (**d**) *Dipterygium glaucum* Decne. (**e**) *Heliotropium kotschyi* Gürke. (**f**) *Tribulus pentandrus* Forssk. (**g**) *Zygophyllum qatarense* Hadidi.

**Table 1 molecules-28-01504-t001:** Free amino acids contents (g/100 g) in the leaves of seven wild species from the United Arab Emirates.

Amino Acid	*Chenopodium murale* L.	*Dipterygium glaucum* Decne.	*Heliotropium digynum* Asch. ex C.Chr.	*Heliotropium kotschyi* Gürke	*Salsola imbricata* Forssk.	*Tribulus pentandrus* Forssk.	*Zygophyllum qatarense* Hadidi	Human Adult Requirements, mg/kg per Day
Alanine	0.103	0.286	0.424	0.12	0.059	0.27	0.07	-
Arginine	0.122	0.433	0.359	0.151	0.144	0.275	0.093	-
Aspartic acid	0.213	0.672	1.35	0.43	0.163	0.588	0.133	-
Valine *	0.099	0.27	0.397	0.14	0.097	0.274	0.062	10
Glutamic acid	0.234	0.702	1.499	0.268		0.762	0.206	-
Glycine	0.099	0.28	0.368	0.118	0.08	0.336	0.065	-
Threonine *	0.092	0.252	0.365	0.117	0.096	0.329	0.066	7
Isoleucine *	0.082	0.208	0.296	0.103	0.049	0.142	0.056	10
Leucine *	0.149	0.385	0.571	0.185	0.083	0.481	0.107	14
Histidine *	0.073	0.134	0.168	0.08	ND	0.100	ND	8–12
Cystine ^+^	0.023	0.041	0.029	ND	ND	0.03	ND	13
Methionine ^+^	0.047	0.067	0.021	0.031	0.02	0.031	0.021	
Proline	0.087	0.245	0.343	0.116	0.076	0.321	0.078	-
Lysine *	0.106	0.287	0.455	0.124	0.064	0.267	0.056	12
Serine	0.09	0.243	0.355	0.115	0.072	0.394	0.051	-
Tyrosine ^#^	0.107	0.178	0.224	0.113	0.135	0.219	0.081	14
Phenylalanine *^,#^	0.134	0.295	0.374	0.132	0.072	0.244	0.088	

Human daily requirements are recommended by WHO, 1985; ^*^—essential amino acids; ^#^—tyrosine + phenylalanine; ^+^—cystine + methionine; ND—not detectable.

**Table 2 molecules-28-01504-t002:** Fatty acid compositions (% of total fat) of the United Arab Emirates wild plants.

Fatty Acid	TP	HD	DG	SI	HI	ZQ	CM
C12:0	ND	ND	ND	ND	0.93	ND	ND
C14:0	2.58	2.15	2.33	0.19	1.18	2.21	3.26
C16:0	32.4	30.7	24.3	15.7	19.8	28.2	32.8
C16:1	ND	1.18	2.03	ND	0.34	1.35	0.43
C18:0	48.5	17.2	16.1	7.61	12.6	16.9	50.4
C17:0	0.6	0.38	ND	0.06	ND	0.12	0.88
C18:1	10.5	23	27.9	51.8	35.5	25.7	8.58
C18:2 *ɯ*6	3.99	15	16.3	24	28.2	15.6	2.8
C18:3 *ɯ*3	ND	8.74	9.88	0.21	0.58	8.99	0.56
C20:0	1.09	1.25	1.16	0.34	0.84	1.18	ND
C20:1	ND	ND	ND	0.13	ND	ND	ND

TP—*Tribulus pentandrus*; HD—*Heliotropium digynum*; DG—*Digiterygium glaucum*; SI—*Salsola imbricata*; HI—*Heliotropium kotschyi*; ZQ—*Zygophyllum qatarense*; CM—*Chenopodium murale*; ND—not detectable.

**Table 3 molecules-28-01504-t003:** Nutritional composition of the different plant species.

Parameter	TP	HD	DG	SI	HI	ZQ	CM WHO	Recommended Rates
Saturated Fats (g/100 g)	0.34	0.31	0.45	0.13	0.11	0.2	0.34	10% of total kcal/day
Mono-Unsaturated Fats (g/100 g)	0.19	0.21	0.21	0.13	0.23	ND	0.19	-
Poly-Unsaturated Fats (g/100 g)	0.17	0.18	0.21	0.10	0.11	ND	0.17	-
Energy (kcal/100 g)	95	98	124	25	57	32	22	18–25 (Kcal/kg bwt.)/day
Fat (g/100 g)	0.7	0.7	0.87	0.36	0.45	0.23	0.76	-
Carbohydrates (total) (g/100 g)	16.5	15.8	22.7	3.7	10	5.8	0.7	130 g/day
Proteins (g/100 g)	5.72	7.21	6.22	1.87	3.2	1.57	3.09	0.75 g/kg/day
Salt (g/100 g)	0.042	0.087	0.071	3.61	0.594	1.08	1	5 g/day
Ash (total) (g/100 g)	6.57	3.9	3.5	7.6	4.1	7.78	6.98	-
Moisture (g/100 g)	70.5	72.4	66.7	86.5	82.2	84.6	88.5	-
Crude Fiber (%)	0.071	0.032	0.152	0.02	0.06	0.042	0.018	-
Ash (Insoluble in acids) (g/100 g)	1.9	1.2	1	2.2	1.2	2.3	2.1	-
Neutral Detergent Fiber (NDF) (%)	13.9	17	20	5.24	10	5.56	3.94	-
Acid Detergent Fiber (ADF) (%)	7.78	10.7	12.2	0.717	5.67	1.89	1.69	-

TP—*Tribulus pentandrus*; HD—*Heliotropium digynum*; DG—*Digiterygium glaucum*; SI—*Salsola imbricata*; HI—*Heliotropium kotschyi*; ZQ—*Zygophyllum qatarense*; CM—*Chenopodium murale*; bwt—body weight; ND—not detectable.

**Table 4 molecules-28-01504-t004:** Composition of vitamin and minerals in the plants.

Parameter	TP	HD	DG	SI	HI	ZQ	CM
Vitamin A (free Retinol) (mg/100 g)	0.02	0.02	ND	ND	ND	ND	0.08
Vitamin B2 (Riboflavin) (mg/Kg)	6.29	ND	1.41	ND	ND	ND	ND
Vitamin B3 (Niacin) (mg/100 g)	0.37	0.095	0.21	0.08	ND	0.09	0.03
Vitamin B6 (Pyridoxin) (mg/100 g)	0.06	0.03	0.10	0.03	ND	0.03	0.03
L-ascorbic acid (Vitamin C) (mg/100 g)	20	20	20	20	20	20	20
Phosphorus (P) (mg/Kg)	1154	502	922	137	280	133	762
Nitrogen (N) (mg/Kg)	9157	11,532	9958	2996	5117	2514	4939
Zinc (Zn) (mg/Kg)	642	87.6	1.92	27.3	70	0.233	36.7
Calcium (Ca) (mg/Kg)	4595	1710	9662	6100	8333	8825	3753
Potassium (K) (mg/Kg)	6665	2050	7476	4395	4509	438	9406

TP—*Tribulus pentandrus*; HD—*Heliotropium digynum*; DG—*Digiterygium glaucum*; SI—*Salsola imbricata*; HI—*Heliotropium kotschyi*; ZQ—*Zygophyllum qatarense*; CM—*Chenopodium murale*; ND—not detectable.

**Table 5 molecules-28-01504-t005:** Information on wild plant species analyzed for their nutritional composition.

Species	Family	Local Name	Life Cycle	Habitat	Native Range	Uses
*Chenopodium murale* L.	Amaranthaceae	Abu’ efei	Annual	Plantation, fallow fields	Europe, N Africa, Arabian Peninsula, SW Asia	Vegetable, fodder
*Dipterygium glaucum* Decne.	Cappraceae	Safrawi	Perennial	Sandplains	NE Africa, Arabian Peninsula, Iran, and S Asia	Fodder, medicine
*Heliotropium digynum* Asch. ex C.Chr.	Boraginaceae	Kary, Jery	Perennial	Sandplains	N Africa, W Asia including Arabian Peninsula	Fodder
*Heliotropium kotschyi* Gürke	Boraginaceae	Ramram	Perennial	Sandplains, gravels	NE Africa, Arabian Peninsula and parts of SW Asia	Medicine
*Salsola imbricata* Forssk	Amaranthaceae	Ghadraf	Perennial	Saline sand, disturbed land	N Africa, Arabian Peninsula and SW Asia	Fodder, medicine
*Tribulus pentandrus* Forssk.	Zygophyllaceae	Shersir	Perennial	Sandplains, valleys	N Africa, SW Asia	Fodder
*Zygophyllum qatarense* Hadidi	Zygophyllaceae	Haram	Perennial	Sandplains, coastal areas	Arabian Peninsula	Fodder

## Data Availability

The data may be shared upon a valid request.

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
