# Peer review of "Proximate Composition and Nutritional Values of Selected Wild Plants of the United Arab Emirates"

_molecules, 2023, doi:10.3390/molecules28031504_

Round 1

Reviewer 1 Report

The manuscript needs to address the following issues:

1. Scientific names should be in italic throughout the manuscript.

2. Any specific reason for selecting the 7 wild plants?

3. What could be the reason for the lack of specific amino acids in some wild plants?

4. Is there any studies using any of the seven plants from a different habitat? If so, was there any variation in the nutritive values? It should be included in the result and discussion section.

5. It would be better to use sub-section numbering for results and discussion and materials and methods sections.

6. More recent references should be incorporated into the manuscript.

7. Extensive language correction is needed for the manuscript.

Author Response

  1. Scientific names should be in italic throughout the manuscript.

    Revised as suggested

  1. Any specific reason for selecting the 7 wild plants?

These are commonly found in UAE and are found in fodder values but not yet studied the nutritional composition of the selected plants

  1. What could be the reason for the lack of specific amino acids in some wild plants?

It may be due to some amino acids distribution profiles being less optimal in the selected plants to detect

  1. Is there any studies using any of the seven plants from a different habitat? If so, was there any variation in the nutritive values? It should be included in the result and discussion section.

Revised based on the suggestion (ref. [50])

  1. It would be better to use sub-section numbering for results and discussion and materials and methods sections.
  2. More recent references should be incorporated into the manuscript.

Revised based on suggestion

  1. Extensive language correction is needed for the manuscript.

Edited the document to meet the appropriate language requirement

Reviewer 2 Report

The article corresponds to a chemical and nutritional characterization of plants and although the authors mention in the conclusion that plants could be used for consumption and reduce hunger, toxicological analyzes are not included.

References need to be updated; a large part of the references included were published before 2015. For example, in the introduction, 15 of the 20 references included were published before 2015.

Author Response

The article corresponds to a chemical and nutritional characterization of plants and although the authors mention in the conclusion that plants could be used for consumption and reduce hunger, toxicological analyzes are not included.

The authors recommended the above view in the light of the nutritional composition analysis, further toxicological studies are required for the effective utilization and to determine the accurate rate for consumption

References need to be updated; a large part of the references included were published before 2015. For example, in the introduction, 15 of the 20 references included were published before 2015.

Updated based on the suggestion

Reviewer 3 Report

The topic of the article is very interesting and actual. But, in my opinion the manuscript needs to be improved before it is published. The Materials and methods section is deficient in information and must be improved. The authors should pay more attention to the journal’s template.

 Introduction

  • The term macronutrients are applied to explain carbohydrates, proteins, and fats” – consider rephrasing;
  • Suggestion: Due to the high frequency of malnourishment amongst the susceptible segment of the population – consider adding some statistical data regarding the prevalence of malnutrition;
  • Suggestion: the authors should emphasize the aim and novelty of their study – for example, Apart from C. murale, no other species were analyzed for their nutritional composition earlier (lines 316-317) is better fitted in this section.

Results 

  • Better subtitle - Results and discussions
  • Tables – data is not expressed as mean±standard deviation –was there only one repetition?
  • The numbers of the tables must be corrected in the text;
  • Table 1 – the last line (Total) – has no values;
  • Tables 3 and 4 – measurement unit for each parameter;
  • Suggestion: the authors should include data regarding the recommended intake levels for humans for the investigated nutrients in order to support their conclusion that these plants are a good source of nutrients.

 Materials and methods

  • Collection and Preparation of Samples – appears twice – the information must be compiled
  • The authors gave many characteristics of the soil – it is not clear how they obtained the data
  • “the plants were air dried under shade” – for how long?
  • The plants were powdered using electrical mortar and pestle – are they analyzed immediately or how are they stored?
  • Amino acid analysis using HPLC – no information given about the HPLC system or the reagents used
  • Trace elemental analysis and proximate composition - no information given about the ICP-AES system
  • What method did the authors use to quantify vitamins?
  • No information about the Statistical Analysis – how many replicas? How is data expressed?

References

  • Are not written according to the journal’s template
  • Some references are before 2000 – suggestion: to replace them with more recent data where is possible

Author Response

The topic of the article is very interesting and actual. But, in my opinion the manuscript needs to be improved before it is published. The Materials and methods section is deficient in information and must be improved. The authors should pay more attention to the journal’s template.

 Introduction

  • The term macronutrients are applied to explain carbohydrates, proteins, and fats” – consider rephrasing;
  • Suggestion: Due to the high frequency of malnourishment amongst the susceptible segment of the population – consider adding some statistical data regarding the prevalence of malnutrition;
  • Suggestion: the authors should emphasize the aim and novelty of their study – for example, Apart from C. murale, no other species were analyzed for their nutritional composition earlier (lines 316-317) is better fitted in this section.

Authors incorporated appropriate references and edited the manuscript based on the suggestions

Results 

  • Better subtitle - Results and discussions
  • Tables – data is not expressed as mean±standard deviation –was there only one repetition?
  • The numbers of the tables must be corrected in the text;
  • Table 1 – the last line (Total) – has no values;
  • Tables 3 and 4 – measurement unit for each parameter;
  • Suggestion: the authors should include data regarding the recommended intake levels for humans for the investigated nutrients in order to support their conclusion that these plants are a good source of nutrients.

The sections are subtitled, the table numbers are updated, removed extra rows in table 1, included the units of each parameter and included the available data of recommended rate of investigated nutrients.

The results are not expressed as mean±standard since the analysis used only one repetition.

 Materials and methods

  • Collection and Preparation of Samples – appears twice – the information must be compiled
  • The authors gave many characteristics of the soil – it is not clear how they obtained the data
  • “the plants were air dried under shade” – for how long?
  • The plants were powdered using electrical mortar and pestle – are they analyzed immediately or how are they stored?
  • Amino acid analysis using HPLC – no information given about the HPLC system or the reagents used
  • Trace elemental analysis and proximate composition - no information given about the ICP-AES system
  • What method did the authors use to quantify vitamins?

Incorporated a supplementary file to explain detail procedures

  • No information about the Statistical Analysis – how many replicas? How is data expressed?

References

  • Are not written according to the journal’s template

  • Some references are before 2000 – suggestion: to replace them with more recent data where is possible

 The manuscript is revised according to all the suggestions and comments from the reviewers.

Round 2

Reviewer 1 Report

The manuscript has been improved as per the comments. No more comments

Author Response

The manuscript is revised based on reviewer comments.

Reviewer 2 Report

The manuscript was improved, its acceptance is recommended

Author Response

(The authors gave the same response as above.)

Reviewer 3 Report

The authors did a good job revising the manuscript. The quality of the paper significantly improved. 

There are some language mistakes that need to be corrected.

Other observations:

  • In Table 1 - Requirements (Essential amino acids) mg/kg per day, by age group - the values are according to WHO, 1985? It is not clear.
  • The biochemical analysis of the leaves of 7 wild species indicates the presence of 11 fatty 297 acids, though they were studied for 22 of them (Table 3). - not Table 2?
  • The proximate composition of selected seven wild plants is described in Table 4. The pre- 333 dominant nutrients in different plants include carbohydrates, proteins, fats, ash, and fiber 334 content. - not Table 3?

Author Response

(The authors gave the same response as above.)
